# A Secure and Lightweight ECC-Based Authentication Protocol for Wireless Medical Sensors Networks

**DOI:** 10.3390/s25216567

**Published:** 2025-10-24

**Authors:** Yu Shang, Junhua Chen, Shenjin Wang, Ya Zhang, Kaixuan Ma

**Affiliations:** School of Mathematics and Computer Science, Yunnan Minzu University, Kunming 650504, China; thornbirds@ymu.edu.cn (Y.S.); 041229@ymu.edu.cn (J.C.); 23213038150004@ymu.edu.cn (S.W.); 22213037550006@ymu.edu.cn (Y.Z.)

**Keywords:** authentication and key agreement (AKA), insider privilege attacks, ESL attack, ECC, WMSNs

## Abstract

Wireless Medical Sensor Networks (WMSNs) collect and transmit patients’ physiological data in real time through various sensors, playing an increasingly important role in intelligent healthcare. Authentication protocols in WMSNs ensure that users can securely access real-time data from sensor nodes. Although many researchers have proposed authentication schemes to resist common attacks, insufficient attention has been paid to insider attacks and ephemeral secret leakage (ESL) attacks. Moreover, existing adversary models still have limitations in accurately characterizing an attacker’s capabilities. To address these issues, this paper extends the traditional adversary model to better reflect practical deployment scenarios, assuming a semi-trusted server and allowing adversaries to obtain users’ temporary secrets. Based on this enhanced model, we design an efficient ECC-based authentication and key agreement protocol that ensures the confidentiality of users’ passwords, biometric data, and long-term private keys during the registration phase, thereby mitigating insider threats. The proposed protocol combines anonymous authentication and elliptic curve cryptography (ECC) key exchange to satisfy security requirements. Performance analysis demonstrates that the proposed protocol achieves lower computational and communication costs compared with existing schemes. Furthermore, the protocol’s security is formally proven under the Random Oracle (ROR) model and verified using the ProVerif tool, confirming its security and reliability. Therefore, the proposed protocol can be effectively applied to secure data transmission and user authentication in wireless medical sensor networks and other IoT environments.

## 1. Introduction

With the advancement of technologies such as wireless communication, low-power integrated circuits, and sensors, WMSNs are driving the medical system toward greater intelligence and real-time responsiveness. An increasing number of smart devices are being integrated into WMSNs to improve the efficiency and accuracy of disease diagnosis, treatment response, vital sign monitoring, and health management [1]. The proposed system model is illustrated in Figure 1. It mainly consists of three types of entities—users $U_{i}$, medical sensors $MS_{j}$ and servers $Svr_{k}$—where 1<i<r, 1<j<s and 1<k<t, with r≫s and t≫s. Here, *r*, *s*, and *t* denote the numbers of users, servers, and medical sensors, respectively. Each remote user $U_{i}$ can securely access medical services and sensor data through one or more servers $Svr_{k}$, which may represent hospital gateways or cloud-based healthcare platforms. Each server is responsible for system initialization and key management within its own domain. In large-scale medical systems, multiple servers can coexist, with each server managing a specific group of users and medical devices. All servers are considered semi-trusted entities that honestly execute the protocol but may attempt to infer private information. Each medical sensor $MS_{j}$ belongs to only one subsystem managed by a specific server and is responsible for collecting physiological data (e.g., heart rate, blood glucose, and blood oxygen) [2] and securely transmitting them to the corresponding server [3].

In WMSNs, besides security concerns, many practical challenges must be addressed to ensure reliable operation. These include signal interference, limited power supply, narrow bandwidth, low storage and computational capability, high data redundancy, and adverse environmental conditions that may affect sensing accuracy and communication quality. Such issues have been widely discussed in recent studies on wireless and medical sensor networks [4,5]. Therefore, any practical WMSNs protocol should not only provide strong security protection but also consider these physical and environmental constraints to maintain reliability and scalability in real deployments.

However, the openness of public communication channels allows unauthorized attackers to intercept, tamper with, or even forge the transmitted data [6]. For example, an attacker could manipulate the frequency settings of a pacemaker or alter the insulin dosage delivered by a pump, thereby posing a serious threat to the patient’s life [7,8]. Therefore, ensuring the secure transmission of medical information not only relies on data encryption, but also requires authentication protocols to verify the identities of communicating parties and to maintain data integrity. Authentication protocols ensure the security of the data transmission process by preventing unauthorized access and modification, thereby protecting both parties in the communication and preserving the confidentiality of sensitive medical data. In addition, the privacy protection of image and physiological data is also a key security challenge in WMSNs. Recent research [9,10] has explored privacy-preserving techniques from the perspective of image compression and anti-forensics. These studies complement the proposed ECC-based three-factor authentication protocol by addressing the confidentiality and imperceptibility of medical image transmission. In future work, integrating such image-level privacy mechanisms could further strengthen the overall security framework of WMSNs.

Many researchers have proposed various authentication protocols for WMSNs [11,12]. Numerous security threats (such as offline dictionary guessing and node capture attacks) and design challenges (such as the trade-off between security and performance) have been revealed. However, only a few studies have paid attention to insider privileged attacks and ESL attacks. In WMSNs, the server plays a vital role. It is not only responsible for verifying identities but also for protecting the security of communication content. However, existing protocols generally treat the server as a fully trusted node, which is not always the case in real-world situations. Although some protocols [13,14] enhance password protection by combining user passwords, biometric information, and random numbers, they still generate the terminal device’s long-term private key on the server side, thereby neglecting the security risks introduced by the semi-trusted server. Other schemes [15,16] adopt the method of generating long-term keys independently at the user and device sides to enhance key privacy, but in the registration phase, the server still holds critical information. As a result, internal personnel of the server can combine this information with the data on the public channel to launch impersonation attacks. Some studies [17,18] focus on the ability of the protocol to resist known security threats (such as synchronization attacks, replay attacks, and offline dictionary guessing attacks), but they fail to prevent server insiders from deriving the session key using registration information. In addition, the ESL attack is also a long-overlooked but potentially severe threat in WMSN environments. The fundamental reason for this vulnerability lies in the fact that many ECC-based authentication protocols, in order to reduce resource consumption, generally adopt lightweight key agreement designs centered on temporary random numbers. Although this improves protocol efficiency, it significantly reduces its security in the case of ephemeral secret leakage. For example, in schemes [19,20], the generation of the session key mainly depends on temporary random numbers and public parameters (such as elliptic curve points *G* or *P*, as well as intermediate variables transmitted through the public channel), which allows an attacker to easily derive the session key once the ephemeral secret is obtained. In summary, insider attacks and ESL attacks are two important issues that have not been sufficiently emphasized in existing research.

To address the above issues, this paper first analyzes internal privileged attacks and categorizes them into three typical scenarios: (1) a registered user infers another user’s password using known information and launches an impersonation attack; (2) the server generates and holds the private key of the terminal device, allowing internal personnel to forge the device’s identity; and (3) internal adversaries derive session keys by combining registration data with messages transmitted over the public channel. To counter these threats, this paper proposes the following measures: the user password is hashed together with biometric factors and random numbers to enhance resistance against guessing attacks; a user-side random number update mechanism is introduced during the registration process; and the server is only responsible for generating part of the terminal’s private key to prevent it from having full control over the device’s key, thereby reducing the risk of internal attacks. Regarding ESL attacks, this paper identifies their root cause as the protocol’s excessive reliance on ephemeral secrets and proposes a key agreement mechanism that combines long-term private keys with temporary random numbers to enhance robustness. In addition, the attacker model is extended. In the current research, the attacker model serves as the foundation for secure protocol design, defining the assumed adversarial capabilities and providing the basis for security analyses. The four-category attacker model proposed by Wang et al. in 2018 [21], widely adopted in the literature, primarily targets external attackers and fails to capture threats such as server corruption and ephemeral secret leakage. Building upon this, the paper introduces attacker capabilities related to internal privilege attacks and ESL attacks, improving the categorization of attacker capabilities and making the security analysis more relevant to the actual environment. Based on this model, we propose a more robust three factor authentication protocol based on ECC. The proposed protocol’s security is formally proven under the ROR model [22], and its correctness is further validated using the automated formal verification tool ProVerif. The results show that the protocol can effectively resist common known attacks while achieving better communication and computational efficiency.

Compared with previous protocols, the main contributions of this paper are as follows:We analyze Wang et al.’s ECC-based protocol [18] and identify two major vulnerabilities—ESL and gateway impersonation attacks.A robust ECC-based three-factor authentication protocol for WMSNs is proposed. It leverages elliptic curve operations for lightweight key generation and introduces an enhanced adversary model covering insider and ESL attacks.The protocol is formally verified under the ROR model and through automated analysis with the ProVerif tool. The findings confirm that the proposed scheme withstands known attacks and satisfies security requirements in WMSNs settings.Compared with current protocols, the proposed protocol provides enhanced security features while improving computational and communication efficiency.

The structure of the paper is as follows: Section 2 provides an overview of related work. Section 3 presents an enhanced adversary model and evaluation criteria. Section 4 highlights the weaknesses in Wang et al.’s protocol [18]. Section 5 provides a detailed explanation of the phases of the proposed protocol. Section 6 presents a security analysis and experimental evaluation. Section 7 compares the proposed protocol with other related protocols. The Section 9 concludes the paper.

## 2. Related Work

In 2016, Jiang et al. [23] proposed a three-factor authentication protocol to promote the development of the WMSNs. Although it was claimed to be robust, Ayub et al. [24] pointed out that the protocol could not resist attacks such as user impersonation and smart card loss, and could not provide clock synchronization. Later, Peralta-Ochoa et al. [25] pointed out that the scheme [24] could not defend against man-in-the-middle and replay attacks. Liu and Chung [26] proposed an authentication scheme for wireless medical sensor network applications, claiming it could resist common attacks. In 2018, Challa et al. [27] found that the scheme [26] was vulnerable to threats such as smart card loss, password guessing, user impersonation, and internal privilege attacks, and improved it by proposing a three-factor user authentication and key agreement protocol. Narwal et al. [28] demonstrated that the scheme had low communication and computation overhead, but Wang et al. [29] pointed out that the scheme [27] still could not prevent smart card loss, offline dictionary attacks, and de-synchronization attacks, and lacked user anonymity and forward security.

Although researchers have proposed many authentication protocols for the WMSNs, internal privilege attacks and ESL attacks have long been underemphasized. Although schemes [23,24,26,27] mentioned internal privilege attacks, they mostly only consider the case where the attacker guesses the password as an internal user, ignoring the possibility that the attacker may have the private key of the participants and even be able to calculate the session key using the data stored and transmitted through public channels. Therefore, these schemes cannot effectively resist internal privilege attacks. In 2018, Dhillon et al. [30] proposed an authentication protocol for WMSNs that considered two types of internal privilege attack scenarios and enhanced protection against internal privilege attacks by not storing sensitive information related to users during the registration phase and encrypting the information transmitted in the channel. The studies by Azrour et al. [31] verified that the scheme [30] could effectively defend against internal privilege attacks. However, Mousavi et al. [32] pointed out that the scheme [30] still could not resist eavesdropping attacks and could not provide reliable authentication. Moreover, attackers could obtain ephemeral secrets in the protocol to compute the session key. For a long time, researchers have overlooked this potential risk, leading to many authentication protocols [23,33] failing to effectively defend against ESL attacks. The protocol proposed by Li et al. [34] could resist ESL attacks, but Koya et al. [35] pointed out that the protocol could not prevent node capture attacks and failed to provide untraceability. Similarly, the protocols proposed by Ryu et al. [36] and Roy et al. [37], which could defend against ESL attacks, also faced security vulnerabilities or excessive overhead issues.

In addition to traditional authentication and encryption schemes, recent studies have explored the use of chaotic systems and memristive neural models to enhance data security in multimodal medical environments. For instance, Gao et al. [38] proposed a three-dimensional memristor-based hyperchaotic map for pseudorandom number generation and multi-image encryption, which ensures high-quality randomness and robustness against statistical attacks. Similarly, Gao et al. [39] introduced a video segment encryption method based on the discrete sinusoidal memristive Rulkov neuron, achieving efficient protection for medical video data. These approaches provide new insights into secure key generation and multimedia data protection, which could inspire future extensions of ECC-based authentication protocols for imaging and video monitoring scenarios in WMSNs.

## 3. Enhanced Attacker Model and Evaluation Criteria

In this section, we present the attacker model along with the evaluation criteria. Table 1 provides a complete list of symbols used in the paper.

### 3.1. Attacker Model

In 2018, Wang et al. [21] proposed a stringent attacker model, as shown in A1∼A4 below. However, the research found that servers should not be regarded as trusted entities, as data breaches and unauthorized port listening events have become increasingly common in recent years [40]. To address such threats, we introduced the A6 attacker capability: A can corrupt the server, eavesdrop, and steal messages received by the server during any operation.

However, during the session key agreement process, users and medical sensor used ephemeral secrets (random numbers), which could be leaked. This occurs because these secrets are produced by external sources that A could manipulate. Furthermore, they are often pre-computed and stored in insecure devices. Consequently, if the secrets are leaked, the session key would also be exposed, and the private keys of the user and medical sensors might also be at risk. To address such threats, we introduce the A7 attacker capability: A has a certain level of ability to guess the random numbers of the participants. A’s capabilities are described as follows:A1:The Dolev–Yao model [41] assumes that A can intercept, modify, delete, or block messages on public communication channels.A2:User passwords are typically easy-to-remember strings that follow a Zipf distribution [42]. A can exhaustively search all elements of the user identity space and password space $|D_{id}\left|\;\times \;\right|D_{pw}|$ offline; and when evaluating non-privacy security, A can obtain $U_{i}$’s identity ID.A3:When evaluating *n* factor security (n=2,3), A can obtain any n−1 factors. However, all *n* factors cannot be obtained simultaneously, as this would constitute a trivial attack [43].A4:A can obtain the previous session keys between $U_{i}$ and $MS_{j}$ [21].A5:A physically captures the medical sensor node with the help of power analysis attacks, and can extract all the stored parameters from the memory [44].A6:A can corrupt $Svr_{k}$, eavesdrop on, and steal the messages received by $Svr_{k}$ during any operation.A7:A has a certain ability to guess the random numbers of the protocol participants.

### 3.2. Evaluation Criteria

We established the evaluation criteria shown in Table 2 by following a widely recognized standard framework [43]. To represent resistance against internal attacks, C11 is introduced. First, during the registration phase, it is required that the long-term keys of the user and the terminal device remain secure, and that the user’s password information cannot be accessed by internal users within the registration center. Second, internal users must not be able to obtain the session key established between the user and the terminal device. In addition, C12 is introduced to address resistance against ESL attacks, further enhancing the security of the authentication protocol.

## 4. Review of Wang et al’s Protocol

In this section, we point out that the protocol proposed by Wang et al. [18] is vulnerable to ESL attacks and cannot resist gateway impersonation attacks.

1.ESL Attack: In Wang et al.’s scheme, once A obtains the session’s temporary information $r_{i}$, they can obtain the session key SK through the following steps.(a)A obtains $M_{{sg}_{1}}=\left\{M_{2},M_{3},M_{4},M_{5}\right\}$ through a public channel.(b)A extracts $M_{11}$ from the message.(c)A calculates the session key $SK=h(M_{2}||M_{11}||r_{i}\cdot M_{11})$.2.Gateway Impersonation Attack: A can generate a replay message that can pass the IoT device verification stage through the following steps:(a)A obtains $M_{{sg}_{2}}=\left\{M_{2},M_{6},M_{7},M_{8}\right\}$ through the public channel.(b)A generates $r_{g}^{*}$ and computes:$X_{s_{j}}=h(x_{G_{k}}||SID_{j})$, $M_{9}=h(X_{S_{j}}\left|\right|M_{2})⊕r_{g}^{*}$, $M_{10}=h(M_{2}\left|\right|M_{9}\left|\right|r_{g}^{*}||SID_{j}\left|\right|X_{S_{j}})$.(c)A sends $M_{{sg}_{3}}=\left\{M_{2},M_{9},M_{10}\right\}$ to $S_{j}$.(d)After receiving the message, $S_{j}$ calculates $r_{g}^{\prime },M_{10}^{\prime }$, and verifies $M_{10}^{\prime }$.In this way, the forged message is validated by $S_{j}$ verification, and A successfully performs a gateway impersonation attack.

## 5. Proposed Protocol

### 5.1. Initialization Phase

In this work, we assume that the server operates within a local network environment, such as a hospital’s internal data center, where communication between medical sensors, gateway nodes, and the server is managed using a controlled and trusted infrastructure. Nevertheless, the proposed protocol can also be extended to remote cloud-based servers with minor adjustments in the communication setup and security assumptions.

We employ public-key encryption, fuzzy verifiers, and honey-word techniques to implement multi-factor security. On the medical sensor side, two elliptic curve point multiplication operations are performed to provide forward secrecy [43].

Before system deployment, $Svr_{k}$ needs to perform the following actions:$Svr_{k}$ selects an elliptic curve E(x,y) over the finite field $F_{P}$, a large prime number *p*, and a point $P\in E_{\left(x,y\right)}\left(F_{P}\right)$ as the base point.$Svr_{k}$ selects rk as the global private key and computes PK=rk·P as the global public key.When using this protocol, the global private key of the server is assumed to remain constant during a specific operational period to ensure consistent authentication for registered users and devices. However, it can be periodically refreshed to improve resistance against key exposure.The function h(·) is selected as a one-way hash function.

Finally, the server securely maintains its private key and publishes the system’s public parameters {E(x,y),p,P,PK,h(·)} before system deployment.

### 5.2. Registration Phase

In this protocol, the elliptic curve cryptographic (ECC) operations play a central role in achieving key agreement and user anonymity. Specifically, each entity’s public–private key pair is generated based on elliptic curve point multiplication, which ensures lightweight computation and high security strength.

The registration phase includes both medical sensor registration and user registration, both of which are completed in a secure manner.

Medical sensor registration phase:$MS_{j}$ provides its identifier $ID_{ms}$, generates a random number *c*, calculates $R_{ms}=c\cdot P$, and after the calculation, sends the registration request $\left\{ID_{ms},R_{ms}\right\}$ to $Svr_{k}$.After receiving the registration request, $Svr_{k}$ first checks whether $ID_{ms}$ is valid and does not already exist in the database. If $ID_{ms}$ already exists, rejects $MS_{j}$’s registration request. Otherwise, $Svr_{k}$ generates a random number *d*, computes $K_{ms}=R_{ms}+d\cdot P$, and calculates $W_{sr}=rk\cdot K_{ms}$. $Svr_{k}$ stores $\left\{ID_{ms},K_{ms},W_{sr}\right\}$ and then sends $\left\{ID_{u},K_{ms},d\right\}$ to $MS_{j}$ via a secure channel.After receiving the message, $MS_{j}$ calculates $k_{ms}=\left(c+d\right)\;mod\;p$, computes $W_{rs}=k_{ms}\cdot PK$, and stores $\left\{ID_{u},K_{ms}\right\}$.

It should be noted that the private key of $MS_{j}$ is generated locally on the $MS_{j}$ side rather than on the $Svr_{k}$ side. This approach avoids the risk of private key leakage caused by $Svr_{k}$ being only semi-trusted. In addition, each sensor node only needs to store a small number of cryptographic parameters, including the public system parameters {E(x,y),p,P,PK,h(·)}, its own private key $k_{ms}$, the corresponding public key $K_{ms}$, and $ID_{u}$. Therefore, the memory requirement for each sensor device is minimal, making the proposed protocol suitable for lightweight medical sensors with limited storage capacity. The above process is shown in Figure 2.

User registration phase:$U_{i}$ inputs $ID_{u}$, $PW_{u}$, and $SD_{u}$, and selects a random number *a*. It then uses the fuzzy extractor $Gen\left(Bio_{u}\right)=\left(\delta_{u},\theta_{u}\right)$ to extract biometric information. $HPW_{u}=h(PW_{u}\left|\right|\delta_{u}\left||a\right)$ and $R_{u}=a\cdot P$ are calculated. Finally, $U_{i}$ sends a registration request $\left\{ID_{u},HPW_{u},R_{u}\right\}$ to $Svr_{k}$ via a secure channel.After receiving the registration request, $Svr_{k}$ generates a random number *b*, and calculates $B_{1}=h(ID_{u}\left||rk||b\right)$, $B_{2}=h(HPW_{u}||ID_{u})⊕B_{1}$, and $K_{u}=R_{u}+b\cdot P$. Here, $B_{1}$ is used to conceal $U_{i}$’s true identity, $B_{2}$ is used to transmit $B_{1}$, and $K_{u}$ is $U_{i}$’s public key. After the calculations, $Svr_{k}$ stores $\left\{ID_{u},Honey-List\right\}$ and securely transmits $\left\{ID_{ms},K_{u},b,B_{2},P,PK\right\}$ to $U_{i}$ via a secure channel.After receiving the message, $SD_{u}$ generates a new random number $a^{\prime }$, and calculates $B_{1}=$$B_{2}⊕h(HPW_{u}||ID_{u})$, $HPW_{u}^{new}=h(PW_{u}\left|\right|\delta_{u}\left|\right|a^{\prime })$,$A_{2}=h(ID_{u}||HPW_{u}^{new}\left|\right|B_{1})\;mod\;n_{0}$, $B_{2}=h(HPW_{u}^{new}||ID_{u})⊕B_{1}$, $k_{u}=\left(a+b\right)\;mod\;p$. Finally, $SD_{u}$ stores $\left\{ID_{ms},K_{u},n_{0},P,PK\right\}$.

The user registration stage is illustrated in Figure 3.

The update of the random number *a* is crucial for defending against insider privilege attacks. In the login phase, both the password and biometric features are used to derive $HPW_{u}$ in order to authenticate $U_{i}$ on $SD_{u}$. However, once the administrator of $Svr_{k}$ obtains the parameters stored in $SD_{u}$ and $U_{i}$’s biometric features, they can infer the $PW_{u}$ from $HPW_{u}$, and use $U_{i}$’s login process to verify whether $HPW_{u}$ and $HPW_{u}^{*}$ are equal, thereby checking the correctness of $PW_{u}$. To prevent this, we update the random number *a* and modify $HPW_{u}$ to $HPW_{u}^{new}$. In addition, $U_{i}$’s private key is generated on the $U_{i}$ side rather than on the $Svr_{k}$ side, thereby avoiding the risk of private key leakage caused by the semi-trusted $Svr_{k}$.

### 5.3. Authentication Phase

When $U_{i}$ needs to access data from $MS_{j}$, the $U_{i}$, $Svr_{k}$, and $MS_{j}$ need to go through the following authentication process. Eventually, $U_{i}$ and $MS_{j}$ will establish a session key for secure communication thereafter. The specific process is shown in Figure 4.

$U_{i}⟶Svr_{k}$: $\left\{S_{1},S_{3},S_{4}\right\}$.$U_{i}$ inputs the identity $ID_{u}^{*}$ and password $PW_{u}^{*}$, and the biometric data $Bio_{u}^{*}$ is stored in $SD_{u}$ through the fuzzy extractor. $SD_{u}$ computes $\delta_{u}^{*}=Rep\left(Bio_{u},\theta_{u}\right)$, $HPW_{u}^{*}=h(PW_{u}^{*}\left|\right|\delta_{u}^{*}\left||a\right)$, $B_{1}^{*}=B_{2}⊕h(HPW_{u}^{*}||ID_{u}^{*})$, $A_{2}^{*}=h(ID_{u}^{*}||HPW_{u}^{*}\left|\right|B_{1}^{*})\;mod\;n_{0}$. Then, it checks if $A_{2}^{*}$ is equal to $A_{2}$ to verify $U_{i}$’s legitimacy. If $A_{2}^{*}\neq A_{2}$, rejects $U_{i}$’s login request. Otherwise, generates a random number $a_{1}$ and a timestamp $T_{1}$, and calculates $S_{1}=a_{1}\cdot K_{u}$, $S_{2}=\left(a_{1}k_{u}\;mod\;q\right)\cdot PK$, $S_{3}=h(S_{1}\left|\right|S_{2})⊕(ID_{u}||ID_{ms})$, $S_{4}=h(B_{1}^{*}||ID_{u}\left|\right|S_{1}\left|\right|S_{2}||ID_{ms})$. Finally, the login request $\left\{S_{1},S_{3},S_{4}\right\}$ is sent to $Svr_{k}$.$Svr_{k}⟶MS_{j}$: $\left\{S_{1},S_{6},S_{7}\right\}$.After receiving the login request, $Svr_{k}$ first checks whether $|T_{1}^{\prime }-T_{1}|<\Delta T$ holds. If it does, it calculates $S_{2}^{\prime }=rk\cdot S_{1}$, $ID_{u}^{\prime }||ID_{ms}^{\prime }=h(S_{1}\left|\right|S_{2}^{\prime })⊕S_{3}$, $S_{4}^{\prime }=h(B_{1}||ID_{u}^{\prime }\left|\right|S_{1}\left|\right|S_{2}^{\prime }||ID_{ms}^{\prime })$, and verifies $S_{4}^{\prime }?=S_{4}$. If $S_{4}^{\prime }\neq S_{4}$, $Svr_{k}$ terminates the request. Otherwise, it generates a timestamp $T_{2}$, calculates $S_{5}=h(W_{sr}\left|\right|T_{2})$, $S_{6}=S_{5}⊕ID_{u}^{\prime }$ and $S_{7}=h(S_{1}\left|\right|S_{5}||ID_{ms}^{\prime }||ID_{u}^{\prime })$. Finally, it sends $\left\{S_{1},S_{6},S_{7}\right\}$ to $MS_{j}$.$MS_{j}⟶Svr_{k}$: $\left\{S_{8},S_{9}\right\}$.After receiving the message, $MS_{j}$ first checks whether $|T_{2}^{\prime }-T_{2}|<\Delta T$ holds. If it does, it calculates $S_{5}^{\prime }=h(W_{rs}\left|\right|T_{2}^{\prime })$, $ID_{u}^{\prime \prime }=S_{5}^{\prime }⊕S_{6}$, $S_{7}^{\prime }=h(S_{1}\left|\right|S_{5}^{\prime }||ID_{ms}||ID_{u}^{\prime \prime })$. Then, it checks whether $S_{7}^{\prime }?=S_{7}$. If $S_{7}^{\prime }\neq S_{7}$, it terminates the session. Otherwise, it generates a random number $c_{1}$, calculates $S_{8}=c_{1}\cdot K_{ms}$, $S=c_{1}k_{ms}\;mod\;q)\cdot S_{1}$, $SK=h(S_{1}||S_{8}||S)$, $S_{9}=h(S_{1}\left|\right|S_{8}||ID_{ms}\left|\right|K_{ms})$. Finally, it sends $\left\{S_{8},S_{9}\right\}$ to $Svr_{k}$.$Svr_{k}⟶U_{i}$: $\left\{S_{8},S_{10}\right\}$.After receiving the message, $Svr_{k}$ calculates $S_{9}^{\prime }=h(S_{1}\left|\right|S_{8}||ID_{ms}^{\prime }\left|\right|K_{ms}^{\prime })$. Then, it checks whether $S_{9}^{\prime }?=S_{9}$. If $S_{9}^{\prime }\neq S_{9}$, it terminates the session. Otherwise, calculates $S_{10}=h(S_{1}\left|\right|S_{2}^{\prime }||ID_{u}^{\prime }||ID_{ms}^{\prime }\left|\right|B_{1}^{\prime }\left|\right|S_{8})$. Finally, it sends {$S_{8},S_{10}$} to $U_{i}$.When $U_{i}$ receives $\left\{S_{8},S_{10}\right\}$, it calculates $S_{10}^{\prime }=h(S_{1}\left|\right|S_{2}||ID_{u}||ID_{ms}\left|\right|B_{1}^{*}\left|\right|S_{8})$. Then, it checks whether $S_{10}^{\prime }?=S_{10}$. If $S_{10}^{\prime }\neq S_{10}$, the session is terminated. Otherwise, this indicates that $Svr_{k}$ has successfully authenticated the $U_{i}$. $U_{i}$ accepts $SK=h(S_{1}||S_{8}||\left(a_{1}k_{u}\;mod\;q\right)\cdot S_{8})$ as the session key shared with $MS_{j}$, and the verification process is successfully completed.

Note that we employ public key encryption, fuzzy verifier, and honeywords technology to achieve multifactor security. Among them, the honeywords mechanism records user login failures and works in conjunction with the fuzzy verifier to resist the offline password-guessing attacks mentioned in the evaluation criteria. However, in practical deployment, the performance of the fuzzy extractor may be affected by environmental noise and the precision of biometric data acquisition, which could introduce a certain degree of instability in key reconstruction. To mitigate this issue, appropriate error-correction mechanisms and biometric signal preprocessing techniques can be employed to ensure the reliability of the extracted keys.

### 5.4. Password Change Phase

In this phase, $U_{i}$ can change the password without needing to interact with $Svr_{k}$.

$U_{i}⟶$$SD_{u}$: $\left\{ID_{u},PW_{u},Bio_{u},PW_{u}^{new}\right\}$.$U_{i}$ initiates a password update request to $SD_{u}$ and submits $\left\{ID_{u},PW_{u},Bio_{u},PW_{u}^{new}\right\}$$SD_{u}$ calculates $\delta_{u}^{*}=Rep\left(Bio_{u},\theta_{u}\right)$, $HPW_{u}^{*}=h(PW_{u}^{*}\left|\right|\delta_{u}^{*}\left||a\right)$, $B_{1}^{*}=B_{2}⊕h(HPW_{u}^{*}||ID_{u}^{*})$, $A_{2}^{*}=h(ID_{u}^{*}||HPW_{u}^{*}\left|\right|B_{1}^{*})\;mod\;n_{0}$. If $A_{2}^{*}\neq A_{2}$, it rejects the request. Otherwise, it calculates $HPW_{u}^{new}=h(PW_{u}^{new}\left|\right|\delta_{u}^{*}\left||a\right)$, $B_{1}^{new}=B_{2}⊕h(HPW_{u}^{new}||ID_{u})$, $A_{2}^{new}=h(ID_{u}||HPW_{u}^{new}\left|\right|B_{1}^{new})\;mod\;n_{0}$. Finally, $B_{1}^{*}$ and $A_{2}^{*}$ are replaced with $B_{1}^{new}$ and $A_{2}^{new}$, completing the password update.

The password change phase is performed locally on $U_{i}$’s device without transmitting the new password over the public channel. During this phase, only hashed or encrypted values derived from the new password are used to update the local authentication parameters. Since the new password is never exposed in plaintext and no sensitive information is exchanged with the server, the local password update process is secure.

### 5.5. Re-Registration Phase

$U_{i}$ with frozen accounts can restore the following services during the re-registration phase.

$U_{i}⟶Svr_{k}$: $\left\{ID_{u},HPW_{u},Re_{reg}request\right\}$.$Svr_{k}⟶U_{i}$: $\left\{A_{2}^{new},B_{2}^{new},PK\right\}$. Upon receiving the re-registration request, $Svr_{k}$ first checks the database for the $ID_{u}$; if not found, the request is rejected. Otherwise, it selects a new random number $b^{new}$, computes $B_{1}^{new}=h(ID_{u}\left||rk|\right|b^{new})$, $B_{2}^{new}=h(HPW_{u}||ID_{u})⊕B_{1}^{new}$, stores $\left\{ID_{u},B_{2}^{new},b^{new}\right\}$, and finally, it sends $\{K_{u},ID_{ms},b^{new},$$B_{2}^{new},P,PK\}$ to $U_{i}$.After receiving the response from $Svr_{k}$, $SD_{u}$ selects a new random number $a^{new}$, performs the calculations according to the registration phase process, and finally stores $\left\{k_{u},ID_{ms},n_{0},P,PK\right\}$.

## 6. Security Analysis

### 6.1. Formal Security Analysis

This section provides the formal security proof of the proposed protocol under the ROR model [22].

We formally prove the security of the proposed protocol under the ROR model, providing a game-based proof ($GM_{0}˘GM_{5}$) with step-by-step probabilistic bounds. Each game transition illustrates how A’s advantage is gradually reduced under well-defined security assumptions.

**Players**: In a three-party protocol P, it contains three participants: $U_{i}$, $Svr_{k}$, and $MS_{j}$. During the protocol execution, *U*, Svr, and MS are instantiated as $U_{i}$, $Svr_{k}$, and $MS_{j}$ respectively. Let *I* refer to the set of protocol instances, with $I^{t}$ being the t-th instance.

**Queries**: These query statements aim to simulate the capabilities of a real A, with the following query types available to A:$Execute(U_{i}^{r},Svr_{k}^{s},MS_{j}^{t}):$ This query simulates a passive attack, which is used by A to obtain the information passed between entities.$Send(I,I_{i}^{t},m):$ In this query, an active attack is simulated as *I*, sends a message *m* to entity $I_{i}^{t}$, and receives a response from $I_{i}^{t}$.$Reveal(I^{t}):$ The query simulates the leakage of an established session key by outputting the session key for $I^{t}$ if it has been created.Corrupt(I): This query simulates A’s ability to corrupt and includes three scenarios:For $I=U_{i}$, A has the ability to obtain two of the three factors, i.e., output $\left\{PW_{u},Bio_{u}\right\}$ or $\left\{PW_{u},SD_{u}\right\}$ or $\left\{Bio_{u},SD_{u}\right\}$.For $I=Svr_{k}$, A can obtain the $RC_{k}$’s private key rk and authentication table $\left\{ID_{u},Honey-List\right\}$, i.e., output $\left\{rk,\left\{ID_{u},Honey-List\right\}\right\}$.For $I=MS_{j}$, A can obtain $MS_{j}$’s private key $k_{ms}$, i.e., output $k_{ms}$.$Test(I_{j}^{t}):$ This query aims to define the semantic security of session keys rather than simulate A’s capabilities. It is restricted to “fresh” sessions and is permitted only once. If $I_{j}^{t}$ lacks a session key or the session is not considered “fresh”, it returns ⊥. Otherwise, a random bit *b* is selected. When b=1, the session key is output; when b=0, a random string of identical length is returned.

In addition, it is also necessary to define Partnering, Freshness, Semantic Security, and the Computational Difficulty Problem.

**Partnering**: Assume sid denotes the session identifier and pid denotes the partner identifier. $U_{i}^{r}$ and $MS_{j}^{t}$ are recognized as partners if and only if: (1) mutual authentication is successfully achieved by both instances; (2) both instances hold the same sid; (3) $U_{i}^{r}$’s pif is $MS_{j}^{t}$ and $MS_{j}^{t}$’s pid is $U_{i}^{r}$.

**Freshness**: The following conditions must be satisfied for an instance *I* to be considered fresh: (1) *I* has been authenticated and holds a session key; (2) A has not conducted a Reveal query targeting *I* or its partner; (3) the $Corrupt(U_{i})$ query has been executed, at most, once; (4) the $Corrupt(MS_{j})$ query has not been executed, or even if $Corrupt(Svr_{k})$ has been executed, A has not performed a Send query.

**Semantic Security**: The session key SK’s security is defined through this notion. During protocol P execution, A can perform a polynomial number of Execute, Send, and Reveal queries, along with a single Test query on a fresh instance. At the game’s conclusion, A needs to guess the bit *b*. A correct guess implies that A has successfully compromised the session key’s semantic security, represented by Pr[Succ(A)]. The advantage gained by A in breaching the protocol’s semantic security is calculated as follows:(1)$$Adv_{A}^{P}=\left[2Pr\left[Succ\right]-1\right]\leq \varepsilon$$

**Elliptic Curve Computational Diffie-Hellman Problem (ECCDHP)**: Given three points, $x_{1},x_{2}$, and *P*, on the elliptic curve $E_{p}$, for a probabilistic polynomial-time adversary A, the computation of $x_{1}x_{2}P$ is considered hard. The advantage $Adv^{ECCDHP}\left(A\right)$ can be neglected for sufficiently small ε:(2)$$\begin{array}{c} Adv^{ECCDHP}\left(A\right)=Pr\left[A\left(x_{1}P,x_{2}P\right)=x_{1}x_{2}P;x_{1},x_{2}\in Z_{p}^{*}\right]<\varepsilon \end{array}$$

**ECCDLP**: On the elliptic curve $E_{p}$, given points *P* and xP, it is hard for a probabilistic polynomial-time A to compute *x*. The advantage $Adv^{ECCDLP}\left(A\right)$ can be neglected for sufficiently small ε:(3)$$\begin{array}{cc} Adv^{ECCDLP}\left(A\right)=Pr[A\left(P,xP\right) & =x;x\in Z_{p}^{*},P\in G]<\varepsilon \end{array}$$

Next, we prove that A’s probability of successfully breaking the protocol is negligible.

**Theorem 1.** 
*Assume A performs, at most, $q_{s}$ Send queries, $q_{e}$ Execute queries, and $q_{h}$ Hash queries within polynomial time. Let $Adv^{ECCDHP}\left(A\right)$ and $Adv^{ECCDLP}\left(A\right)$ represent A’s advantage in breaking the ECCDHP and ECCDLP problems, respectively, and let l denote the length of the security parameter. In this case, the advantage of A in compromising protocol P is as follows:*

(4)
$$\begin{array}{c} Adv_{A}^{\mathrm{AKA}}\leq \frac{2q_{h}^{2}+q_{s}}{2^{l}}+\frac{{\left(q_{s}+q_{e}\right)}^{2}}{p}+2q_{hMAX}\left\{Adv^{ECCDHP}\left(A\right),Adv^{ECCDLP}\left(A\right)\right\} \end{array}$$



**Proof.** We prove Theorem 1 through a series of games $\left(Game\;GM_{0}-Game\;GM_{5}\right)$. Let $Succ_{i}$ be the event where A successfully guesses the bit *b* in the Test query of game $GM_{i}$. The advantage in winning game $GM_{i}$ is $Adv_{A,GM_{\rangle }}^{\mathrm{AKA}}$. Hence, $Game\;GM_{0}$ corresponds to the real protocol attack. □

**$Game\;GM_{0}$**: Simulates the real attack in the random oracle model, so we have the following:(5)$$Adv_{A}^{\mathrm{AKA}}=\left[2Pr\left[Succ_{0}\right]-1\right]$$

Intuitive explanation for $GM_{0}\to GM_{1}$: In Game $GM_{1}$, we maintain the real execution of the proposed protocol but limit the adversary to the passive observation of transmitted messages through the Execute oracle. No additional oracles or capabilities are introduced in this step. Therefore, the adversary’s view in $GM_{1}$ is identical to that in $GM_{0}$, meaning that no effective advantage is gained. Hence, we have $Pr\left[Succ_{1}\right]=Pr\left[Succ_{0}\right]$.

**$Game\;GM_{1}$**: In this game, A intercepts the messages between the three participants through the query $Execute(U_{i}^{r},Svr_{k}^{s},MS_{j}^{t})$, and then A can use the Reveal query and the Test query to determine whether the session key SK is real or random. A replaces the real session key with a random value. This modification does not affect the adversary’s advantage because the key is computationally indistinguishable under the hardness of the ECDLP problem, meaning that advantage does not increase compared to $Game\;GM_{0}$:(6)$$Pr\left[Succ_{1}\right]=Pr\left[Succ_{0}\right]$$

The intuitive explanation for $GM_{1}\to GM_{2}$ is as follows: In this step, the adversary is allowed to actively send forged messages using the Send oracle. However, these forgeries can only be accepted if rare events, such as hash collisions or nonce collisions, occur. Therefore, the difference in advantage between $GM_{1}$ and $GM_{2}$ is negligible and bounded by the probability of these collision events.

**$Game\;GM_{2}$**: In this game, by adding Send queries and Hash queries, we can transform $Game\;GM_{1}$ into $Game\;GM_{2}$, where A constructs a forged message that is believed by the real communication parties. The protocol’s semantic security is breached only when A discovers a collision that results in a valid message. Our protocol features two types of collisions:The occurrence of a collision in the output of the hash function, with a probability no greater than $\frac{q_{h}^{2}}{2^{l+1}}$;The occurrence of a collision in the random number $a_{1}$, with a probability no greater than $\frac{{\left(q_{s}+q_{e}\right)}^{2}}{2p}$.

Therefore, unless one of the above two collisions occurs, A’s advantage remains the same as in game $Game\;GM_{1}$. We have the following:(7)$$|Pr\left[Succ_{2}\right]-Pr\left[Succ_{1}\right]|\leq \frac{q_{h}^{2}}{2^{l+1}}+\frac{{\left(q_{s}+q_{e}\right)}^{2}}{2p}$$

Intuitive Explanation for $GM_{2}\to GM_{3}$: In Game $GM_{3}$, we terminate the simulation when the adversary correctly guesses the verification values (such as $S_{4}$, $S_{7}$, $S_{9}$, or $S_{10}$) without querying the corresponding hash oracles. This step models the event that an adversary forges valid authentication tags by random guessing. The success probability in such a case is negligible and bounded by $\frac{1}{2^{l}}$.

**$Game\;GM_{3}$**: In this game, A replaces certain intermediate parameters ($S_{4},S_{7},S_{9},S_{10}$) related to $Svr_{k}$’s or $U_{i}$’s secret values with randomly generated ones. Due to the security of the ECC-based key derivation, the adversary gains no additional information from this substitution. Thus, we have the following:(8)$$|Pr\left[Succ_{3}\right]-Pr\left[Succ_{2}\right]|\leq \frac{q_{s}}{2^{l}}$$

Intuitive Explanation for $GM_{3}\to GM_{4}$: In this step, we idealize the computation of the session key components, assuming that the adversary cannot derive them without solving the elliptic curve computational problems (ECCDHP and ECCDLP). This transformation represents the reduction from the protocol’s real security to the hardness of standard ECC problems. Thus, the difference in the adversary’s advantage between $GM_{3}$ and $GM_{4}$ is bounded by the advantage of solving these problems.

**$Game\;GM_{4}$**: This game considers the security of the session key. Since SK contains $\left(a_{1},k_{u}\right)$ and $\left(c_{1},k_{ms}\right)$, A cannot know the correct values without the corresponding long-term and short-term secrets. A can use Execute and Hash queries to compute the parameters. There are the following four cases:A executes $Corrupt(U_{i})$ and $Corrupt(MS_{j})$, meaning A can obtain $k_{u}$ and $k_{ms}$ for $U_{i}$ and $MS_{j}$ but cannot obtain the ephemeral secret.A executes $Reveal(U_{i})$ and $Corrupt(MS_{j})$. In this case, A can obtain the ephemeral secret $a_{1}$ of $U_{i}$ and the long-term secret $k_{ms}$ of $MS_{j}$.A executes $Corrupt(U_{i})$ and $Reveal(MS_{j})$. In this case, A can obtain the long-term secret $k_{u}$ of $U_{i}$ and the ephemeral secret $c_{1}$ of $MS_{j}$.A executes $Reveal(U_{i})$ and $Reveal(MS_{j})$, meaning A can obtain the ephemeral secret $a_{1}$ of $U_{i}$ and $c_{1}$ of $MS_{j}$.

In the above four cases, without solving ECCDHP or ECCDLP, A is unable to derive the session key SK. Therefore, $Game\;GM_{4}$ and $Game\;GM_{3}$ are indistinguishable as long as ECCDHP and ECCDLP remain consistent. Thus, we have the following:(9)$$\begin{array}{c} |Pr\left[Succ_{4}\right]-Pr\left[Succ_{3}\right]|\leq_{hMAX}\left\{Adv^{ECCDHP}\left(A\right),Adv^{ECCDLP}\left(A\right)\right\} \end{array}$$

Intuitive Explanation for $GM_{4}\to GM_{5}$: Finally, in Game $GM_{5}$, the real session key is replaced with a truly random key of the same length. The adversary’s goal now becomes distinguishing the real key from the random one. Since all previous events have negligible probabilities, the adversary’s advantage in this final game is approximately zero, meaning that the proposed protocol is secure under the ROR model.

**$Game\;GM_{5}$**: Compared to $Game\;GM_{4}$, this game simulates the situation where A executes $SK=h(S_{1}||S_{8}||S)$ query. If A sends a Test query, the game will be aborted. We can conclude that(10)$$|Pr\left[Succ_{5}\right]-Pr\left[Succ_{4}\right]|\leq \frac{q_{h}^{2}}{2^{l+1}}$$

After all the oracles have been completed, A needs to distinguish between the random value and the actual session key. In the Test query, A has a $\frac{1}{2}$ chance of obtaining the correct key parameter. Therefore, we can conclude that(11)$$Pr\left[Succ_{5}\right]=\frac{1}{2}$$

By considering all possibilities, we prove that Theorem1 holds.

### 6.2. Descriptive Security Analysis

Session Key Agreement: After the mutual authentication is completed, $U_{i}$ and $MS_{j}$ share a session key $SK=h(S_{1}\left|\right|S_{9}||S)=h(S_{1}\left|\right|S_{9}\left|\right|a_{1}k_{u}\cdot S_{9})=h(a_{1}\cdot K_{u}\left|\right|c_{1}\cdot K_{s}\left|\right|a_{1}k_{u}c_{1}\cdot K_{s})$, which is used to protect subsequent communication between $U_{i}$ and $MS_{j}$. Since the random numbers $a_{1}$ and $c_{1}$ are unique for each session, each session key is independent of the others. Therefore, the exposure of the session key in one session does not influence the keys established previously or in the future.Mutual Authentication: $U_{i}$ and $MS_{j}$ achieve mutual authentication through $Svr_{k}$. Specifically, $U_{i}$ and $Svr_{k}$ authenticate each other by verifying whether $S_{4}^{\prime }?=S_{4}$ and $S_{10}^{\prime }?=S_{10}$ hold. Similarly, $Svr_{k}$ and $MS_{j}$ achieve mutual authentication by verifying whether $S_{7}^{\prime }?=S_{7}$ and $S_{9}^{\prime }?=S_{9}$ hold. If any of these conditions are not satisfied, the session is terminated. Therefore, the proposed protocol successfully achieves mutual authentication among the three parties.Anonymity and Untraceability: The protocol uses the secret parameter $S_{2}$, generated through public-key technology, to protect $ID_{u}$ and $ID_{ms}$, with $S_{2}$ being different for each session. Specifically, the identity identifiers $ID_{u}$ and $ID_{ms}$ are not directly transmitted to the $Svr_{k}$. Instead, they are sent in the form of $S_{3}=h(S_{1}\left|\right|S_{2})⊕(ID_{u}||ID_{ms})$. The only entities that can calculate $S_{2}$ are $U_{i}$ and $Svr_{k}$, which holds the private key. A cannot obtain $ID_{u}$ and $ID_{ms}$, ensuring the anonymity of $U_{i}$ and $MS_{j}$. On the other hand, since $S_{1}S_{3}S_{4}$ and $S_{5}$ in the login request dynamically change with the random number $a_{1}$, A cannot track a specific $U_{i}$ and $MS_{j}$ by eavesdropping on the login request message.Resistance Smart Device Loss Attack: Assume the $U_{i}$’s smart device $SD_{u}$ is lost and obtained by A, who can retrieve data $\left(ID_{ms},K_{u},n_{0},P,PK\right)$. On the one hand, if A wants to change the password without being noticed by the device, they must construct the correct $A_{2}^{*}=h(ID_{u}^{*}||HPW_{u}^{*}\left|\right|B_{1}^{*})\;mod\;n_{0}$ in order to pass the verification. However, the data retrieved by A does not help in computing $A_{2}^{*}$. On the other hand, if A wants to correctly guess the password, they can use $A_{2}^{*}$ and $S_{4}$ to verify the correctness of their guess. For $A_{2}^{*}$, even if A with biometric features finds identity and password that satisfy $h(ID_{u}^{*}||HPW_{u}^{*}\left|\right|B_{1}^{*})\;mod\;n_{0}=A_{2}^{*}$, in order to further confirm the password’s correctness, A must perform online verification, which will be blocked by the Honey−List. For $S_{4}$, as described in (3), only the real $U_{i}$ who selects $a_{1}$ and the $Svr_{k}$ that knows the private key rk can compute $S_{2}$. A cannot compute $S_{2}$, and therefore cannot construct $S_{4}$, making it impossible to guess the password’s correctness by comparing $S_{4}^{\prime }$ and $S_{4}$. In summary, our scheme is resilient to such attacks.Resistance User Impersonation Attack: A impersonates the $U_{i}$ by forging the login request $S_{1},S_{3},S_{4}$, where $S_{4}$ is composed of $S_{2}$, as discussed in (4). A cannot compute $S_{2}$. Therefore, the proposed protocol is capable of defending against user impersonation attacks.Resist De-Synchronization Attack: In our protocol, we use random numbers and public-key algorithms to achieve user anonymity and resist replay attacks. Participants do not need to maintain clock synchronization consistency or some temporary certificate-related parameters. Therefore, our scheme can resist de-synchronization attacks.Resistance Replay Attack: Suppose A has obtained all the login and authentication messages transmitted through a public channel and attempts to replay them to $U_{i}$, $Svr_{k}$, and $MS_{j}$. However, in each session, new random numbers $a_{1},c_{1}$ and timestamps $T_{1}$, $T_{2}$ are generated. Once the replayed messages reach $Svr_{k}$ and $MS_{j}$, both entities will verify $S_{4}^{\prime }=h(B_{1}||ID_{u}^{\prime }\left|\right|S_{1}\left|\right|S_{2}^{\prime }||ID_{ms}^{\prime }\left|\right|T_{1}^{\prime })$ and $S_{7}^{\prime }=h(S_{1}\left|\right|S_{5}^{\prime }||ID_{ms}||ID_{u}^{\prime })$. Therefore, the proposed scheme can resist replay attacks.Resistance Offline Dictionary Guessing Attack: A can retrieve the parameters from $SD_{u}$ and generate an authentication factor using guessed identity and password. By comparing the generated factor with the real one, A can verify the accuracy of the guessed password. On the one hand, the password is protected by the ‘fuzzy−verifier’ technique, and the ‘honey−list’ limits A’s online guessing attempts by recording failed logins. On the other hand, for the authentication factors transmitted over a public channel, in order to conduct an offline dictionary guess attack, A must compute $S_{4}=h(B_{1}^{*}||ID_{u}\left|\right|S_{1}\left|\right|S_{2}||ID_{ms}\left|\right|T_{1})$. However, only $Svr_{k}$’s private key rk and the $U_{i}$’s private key $k_{u}$ can compute the parameters $B_{1}^{*}$ and $S_{2}$. Therefore, the proposed scheme not only resists offline dictionary guess attacks against smart devices but also against offline dictionary guess attacks over public channels.Perfect Forward Secrecy: As described in (1), the session key $SK=h(S_{1}||S_{8}||S)$ shared between $U_{i}$ and $MS_{j}$ is associated with $U_{i}$’s private key $k_{u}$ and the random numbers $a_{1}$ and $c_{1}$. Even if the private key is compromised, A cannot use it to decrypt past session records. Since A needs to solve the elliptic curve discrete logarithm problem to obtain the parameter *S*, the future session keys SK remain secure. Therefore, the proposed scheme achieves perfect forward secrecy.Resistance Sensor Node Capture Attack: Assuming A captures $MS_{j}$ and uses power analysis attacks to extract the stored parameters $\left\{ID_{u},K_{ms},P,PK\right\}$, during the authentication phase, $MS_{j}$ sends $\left\{S_{8},S_{9}\right\}$ to $Svr_{k}$, where $S_{8}=c_{1}\cdot K_{ms}$ and $S_{9}=h(S_{1}\left|\right|S_{8}||ID_{ms}\left|\right|K_{ms})$. The parameter $S_{8}$ is generated using the node’s public key $K_{ms}$ and a random number $c_{1}$, which is created by the node itself. A only has $K_{ms}$ and cannot compute $\left\{S_{8},S_{9}\right\}$. Moreover, when A steals the session key, generating $SK=h(S_{1}||S_{8}||S)$ requires calculating $S_{8}$ and *S*, where $S=\left(c_{1}k_{ms}\;mod\;q\right)\cdot S_{1}$. Since the session key SK can only be computed by $U_{i}$ and $MS_{j}$, the proposed scheme effectively resists node capture attacks.Resistance Insider Privilege Attack:After successful registration, A gains access to the registered smart device $SD_{u}$ and extracts the stored data. However, upon receiving the message from the server, the smart device selects a new random number $a^{\prime }$ and updates $HPW_{u}$ to a new value $HPW_{u}^{new}$, ensuring that $HPW_{u}^{new}\neq HPW_{u}$. As a result, $Svr_{k}$ is unable to obtain the verification parameters needed to guess $U_{i}$’s password.The private keys on both the $U_{i}$ side and $MS_{j}$ side are generated locally rather than on the $Svr_{k}$ side, thereby avoiding security threats caused by private key leakage due to the semi-trusted nature of $Svr_{k}$.Assuming that A can obtain the secret parameters transmitted during the registration phase, namely $ID_{u},HPW_{u},R_{U}$ and $ID_{ms},K_{u},b,B_{2},P,PK$, as well as the public channel messages $M_{1},M_{2}$ and $M_{3}$, A is still unable to compute the session key SK. Therefore, the proposed scheme is resistant to internal privileged attacks.Resistance Ephemeral Secret Leakage Attack: The session key $SK=h(S_{1}||S_{8}||S)$, associated with $U_{i}$’s private key $k_{u}$ and the random numbers $a_{1}$ and $c_{1}$, where $S_{1}=a_{1}\cdot K_{u}$, $S_{8}=c_{1}\cdot K_{ms}$ and $S=\left(c_{1}k_{ms}\;mod\;q\right)\cdot S_{1}$. Even if A gains the random numbers $a_{1}$ and $c_{1}$, they cannot compute the $U_{i}$’s private key $k_{u}$. To calculate the session key, A must obtain both the random numbers $a_{1},c_{1}$, and the $U_{i}$’s private key $k_{u}$, which is an infeasible task. Therefore, even if the two random numbers are leaked, the previous session keys will not be compromised.Resistance to Combined and Multi-Adversary Attacks: In addition to insider privileged and ESL attacks, the proposed protocol can also resist more complex threat scenarios involving multiple adversaries or combined attacks. Even if several malicious entities attempt to cooperate, the use of independent session keys and dynamic pseudonyms ensures that compromising one node does not reveal information about others. Furthermore, the mutual authentication and key agreement steps rely on fresh random values and ECC-based computations, preventing coordinated replay or collusion-based attacks.

### 6.3. Automatic Formal Verification by ProVerif

ProVerif [45] is an automated tool for verifying the security of cryptographic protocols. It uses Pi calculus and Prolog-based rules to evaluate protocol confidentiality and supports a wide range of cryptographic primitives, including Diffie–Hellman key exchange, hash functions, and symmetric as well as asymmetric encryption algorithms.

In this section, we employ ProVerif to verify the security of the proposed protocol. The code of the user process is shown in Figure 5, which illustrates the workflow of the login and authentication phases. Similarly, the modeling procedures for $Svr_{k}$ and $MS_{j}$ follow the same approach. In the experiment, sch1 represents the secure channel used for communication between $U_{i}$ and $Svr_{k}$ during the registration phase, while ch1 denotes the public channel used in the authentication phase. In addition, $U_{i}$, $Svr_{k}$ and $MS_{j}$ each have their own processes, represented as (!User)|(!RegenAuth)|(!Device).

We used the standard ProVerify query for verification. The queries were as follows:query attacker(kU).query attacker(rk).query attacker(PWu).query attacker(SKU).query attacker(SKMS).

These queries were respectively used to check whether $U_{i}$’s private key $k_{U}$, $Svr_{k}$’s private key rk, $U_{i}$’s $PW_{u}$, and the session key $SK_{U}$ and $SK_{MS}$ can be derived by the attacker.

The ProVerif tool results are illustrated in Figure 6.

The results indicate that A cannot interfere with the authentication process or retrieve the session key.

## 7. Performance Comparison

This section presents a comparison of our protocol’s performance with that of related protocols [46,47,48,49,50], emphasizing security features, computational efficiency, and communication overhead.

### 7.1. Performance Comparison

The proposed protocol is compared with six other protocols [46,47,48,49,50] regarding functionality and security, as shown in Table 3. Compared with existing protocols, our proposed scheme offers better security and more features.

### 7.2. Computation Cost

For evaluation purposes, we denote $T_{m}$ as the time for a single elliptic curve point multiplication, $T_{h}$ for a one-way hash operation, and $T_{B}$ for fuzzy biometric extraction. The experiments were conducted on two platforms: a server equipped with an Intel(R)Core(TM)i5 2 GHz CPU, 16 GB RAM, and macOS13.4.1, and a RaspberryPI 3 ModelB+ sensor node featuring an ARMCortex−A1.4 GHz processor and 1 GB RAM. For the Curve 25,519 elliptic curve with a 384-bit point length and prime $p=2^{192}$, the average execution times on the server were 1.258 ms for point multiplication, 0.005 ms for hash computation, and 1.258 ms for fuzzy biometric extraction. On the RaspberryPI 3 ModelB+ sensor node, under the same conditions, the corresponding average times were 2.225 ms, 0.019 ms, and 2.225 ms, respectively. Table 4 and Figure 7 shows the time overhead for each authentication and key agreement process in the various protocols.

During the login and authentication phases, $U_{i}$ performs seven one-way hash operations ($T_{h}$) to generate $HPW_{u}^{*}$, $B_{1}^{*}$, $A_{2}^{*}$, $S_{3}$, $S_{4}$, $S_{10}^{\prime }$ and the session key SK. In addition, $U_{i}$ performs three elliptic curve point multiplications ($T_{m}$) for the computations of $S_{1}$, $S_{2}$ and SK, as well as one fuzzy feature extraction operation ($T_{B}$) for biometric verification. Similarly, $Svr_{k}$ executes six one-way hash operations and one point multiplication during the authentication phase, while $MS_{j}$ executes four one-way hash operations and two point multiplications. Therefore, the computational cost of the proposed protocol is expressed as $U_{i}$: $7T_{h}+3T_{m}+T_{B}$, $Svr_{k}$: $6T_{h}+T_{m}$, $MS_{j}$: $4T_{h}+2T_{m}$. So, the total computational cost can be written as $17T_{h}+6T_{m}+T_{B}\approx 8.891$ ms. For comparison, the computational costs of the related protocols were calculated using the same method, among which Huang et al.’s protocol [48] showed the highest computational cost, with a total of $48T_{h}+12T_{m}+T_{B}\approx 16.594$ ms. Li et al.’s protocol [46] showed the lowest computational cost, with a total of $19T_{h}+6T_{m}+T_{B}\approx 8.901$ ms. Compared to Huang et al.’s and Li et al.’s protocol, the computation cost of the proposed protocol are reduced by 46.42% and 0.11%, respectively. When compared with other related protocols [47,49,50], costs reduced by 46.2%, 30.54% and 46.18%, respectively.

Specifically, the computational cost of the intermediate server is reduced by 0.23%, 66.36%, 74.89%, 66.98%, and 66.27%, respectively, as shown in Table 5. These results demonstrate that the proposed scheme optimizes resource consumption and maintains high computational efficiency in resource-constrained medical IoT environments, particularly in scenarios where intermediate servers are connected to multiple sensor nodes.

It is evident that the proposed protocol results in lower computational overhead during the authentication phase.

### 7.3. Communication Cost

Assume the sizes of user identity $ID_{u}$, medical sensor $ID_{ms}$, random number *R*, timestamp *T*, ECC point multiplication *M* and hash output *H* are 64 bits, 64 bits, 256 bits, 32 bits, 320 bits, and 256 bits, respectively. Table 6 and Figure 8 shows the communication overhead for each authentication process in various protocols.

In the proposed protocol, four messages—$M_{1}$, $M_{2}$, $M_{3}$, and $M_{4}$—are exchanged among $U_{i}$, $Svr_{k}$, and $MS_{j}$. Message $M_{1}$ contains $S_{1}$, $S_{3}$, and $S_{4}$, where $S_{1}$ carries a random number, $S_{3}$ transmits the identities of $U_{i}$ and $MS_{j}$, and $S_{4}$ carries a hash value. Thus, the communication cost of $M_{1}$ is M+2ID+H=704 bits. Similarly, the communication costs of $M_{2}$, $M_{3}$ and $M_{4}$ are M+ID+H=640 bits, M+H=576 bits, and M+H=576 bits, respectively. Hence, the total communication cost of the proposed protocol is 704+640+576+576=2496 bits. The communication costs of the related protocols [46,47,48,49,50] were calculated using the same parameter definitions, ensuring consistent evaluation criteria among all the compared schemes. Their total communication overheads are 3200 bits, 2656 bits, 4800 bits, 2816 bits, and 2880 bits, respectively. Compared to these, the total communication overhead of the proposed protocol is reduced by 22%, 6.1%, 48.85%, 11.36%, and 13.33%, respectively.

In addition to achieving lower computational and communication costs compared with existing protocols, further optimization of computational efficiency remains a promising direction. Future work will explore potential improvements, such as lightweight cryptographic primitives, algorithmic refinements, and hardware-assisted acceleration, to further reduce execution time and energy consumption, especially in large-scale WMSNs deployments.

## 8. Experimental Study

### 8.1. Simulation of Secure Communication in a Simplified WMSN Model

To evaluate the practical feasibility of the proposed ECC-based lightweight authentication protocol, a simulation experiment was conducted on a LenovoThinkBook14G7+ laptop using MATLABR2024b. The system was configured with an IntelCorei7 processor, 16 GB RAM, and Windows 11 (64-bit) operating system.

In this experiment, a simplified elliptic curve model was adopted to simulate the authentication and secure communication among three entities, namely the User, Server, and Medical Sensor, within a WMSN. The simulation implemented the elliptic curve defined as: $E:\;y^{2}=x^{3}-x+188\;\left(mod\;751\right)$, where the prime modulus p=751 and the base point *P* = (0.376).

In the MATLAB simulation phase, a smaller modulus was selected to simplify computation and improve execution efficiency. This setting allows for the correctness of the proposed protocol to be verified without affecting its general applicability. This setup validates the correctness of the cryptographic operations without affecting the generality of the protocol.

In this simplified scenario, one user, one server, and one medical sensor were simulated. Each entity independently generated its own elliptic curve key pair, and two rounds of shared key establishment were performed: between the user and the server, and between the server and the sensor. Subsequently, a sample physiological message “Heart rate = 82 bpm” was encrypted and decrypted using the established session key to verify secure data transmission. The partial code implemented in MATLAB is shown in Figure 9. The result is shown in Figure 10.

The results demonstrated that the proposed protocol successfully established session keys and correctly performed the encryption and decryption of medical data, confirming its feasibility in a simplified WMSN environment.

### 8.2. Embedded Implementation and Communication Verification

In order to further verify the feasibility of the proposed authentication protocol in resource-constrained environments, a hardware-based experimental platform was established, as shown in Figure 11. The implementation was carried out on an STM32F103ZET6 microcontroller, which integrates an ARM 32-bit Cortex−M3MCU (STM32F103ZET6, STMicroelectronics, Geneva, Switzerland) running at 72 MHz, with 512 KB Flash memory and 64 KB SRAM. The ECC-based authentication and key-agreement procedures were implemented in C language and executed locally on the STM32 device. Wireless communication between the sensor node and the server was achieved through an ESP-12F Wi-Fi module, enabling encrypted data transmission.

For the implementation of standard ECC operations, we employed Micro ECC [51], a fast and lightweight open-source library supporting elliptic curve cryptographic computations. In the selection of elliptic curves, SECP curves (including Secp160r1, Secp192r1, and Secp256k1) were adopted to evaluate the security and scalability of the proposed protocol under different computational strengths.

The server side was implemented on a laptop equipped with an IntelCore i7-9750H (2.60 GHz) processor, 32 GB RAM, and Windows 10 operating system. The user terminal was simulated on another laptop with an IntelCore i5-8300H (2.30 GHz) processor, 16 GB RAM, and Windows 11. In this setup, the user sends an authentication request, the medical sensor node performs ECC-based key generation and message authentication, and the server validates the request and completes the session establishment.

This experiment corresponds to the communication model presented in Figure 1, where the user, medical sensor node, and server interact through secure channels. The key experimental parameters are listed in Table 7.

## 9. Conclusions

This paper reviews and analyzes the existing authentication protocols in WMSNs, and identifies that current research generally overlooks critical threats such as internal privileged attacks and ESL attacks. A security analysis of the protocol proposed by Wang et al. [18] reveals that it fails to resist ESL attacks and gateway impersonation attacks. To address these issues, this paper proposes a three-factor authentication protocol based on Elliptic Curve Cryptography. The proposed protocol enables secure registration, mutual authentication, and key agreement between mobile devices and sensor nodes over public channels, thereby enhancing both the security and flexibility of the protocol. Furthermore, the protocol’s security is formally proven under the ROR, and its correctness is automatically verified using the ProVerif tool. The results demonstrate that the protocol can effectively defend against multiple threats, including ESL attacks and gateway impersonation attacks. Performance evaluation shows that, compared with some existing schemes, the proposed protocol is more lightweight in terms of computation and communication overhead, achieving improved overall efficiency while ensuring security in resource-constrained WMSNs environments. The proposed protocol can be effectively applied to secure data transmission and user authentication in wireless medical sensor networks and other IoT environments.

With the rapid advancement of quantum computing, traditional cryptographic schemes, including those widely deployed in IoT authentication and key agreement systems, may encounter serious security challenges. Quantum algorithms such as Shor’s algorithm have the potential to compromise classical cryptographic primitives like RSA and ECC, thus threatening the security foundation of current IoT communications. Future research should focus on integrating quantum-resistant cryptographic techniques, such as lattice-based or code-based cryptography, into IoT authentication protocols to ensure long-term resilience in the post-quantum era. In addition, the incorporation of blockchain technology can further enhance decentralized trust management and data integrity in IoT-based medical systems. Moreover, exploring the combination of machine learning or artificial intelligence technologies may enable dynamic adaptation to diverse attack strategies and improve the overall security intelligence of wireless medical sensor networks.

## Figures and Tables

**Figure 1 sensors-25-06567-f001:**
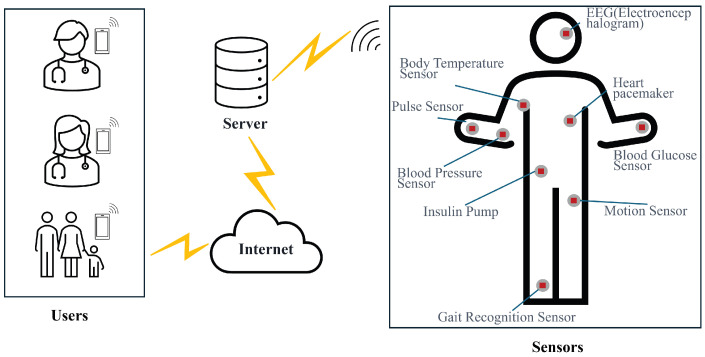
System model of WMSNs.

**Figure 2 sensors-25-06567-f002:**
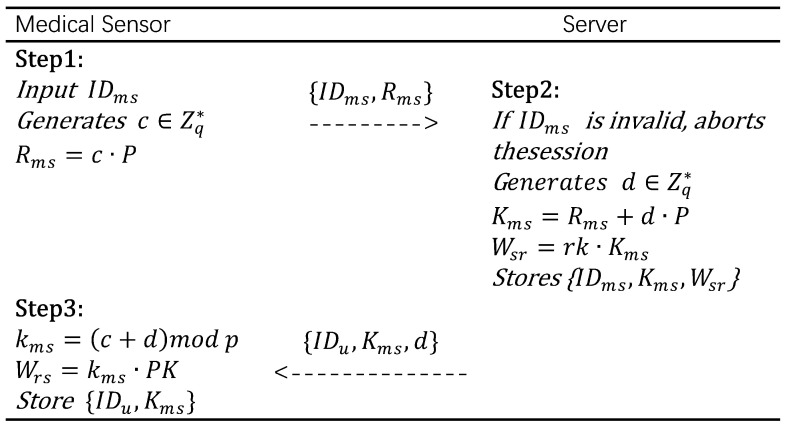
Medical sensor registration phase.

**Figure 3 sensors-25-06567-f003:**
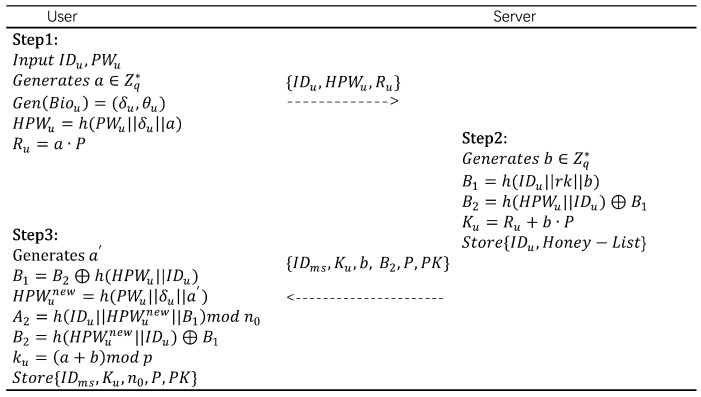
User registration phase.

**Figure 4 sensors-25-06567-f004:**
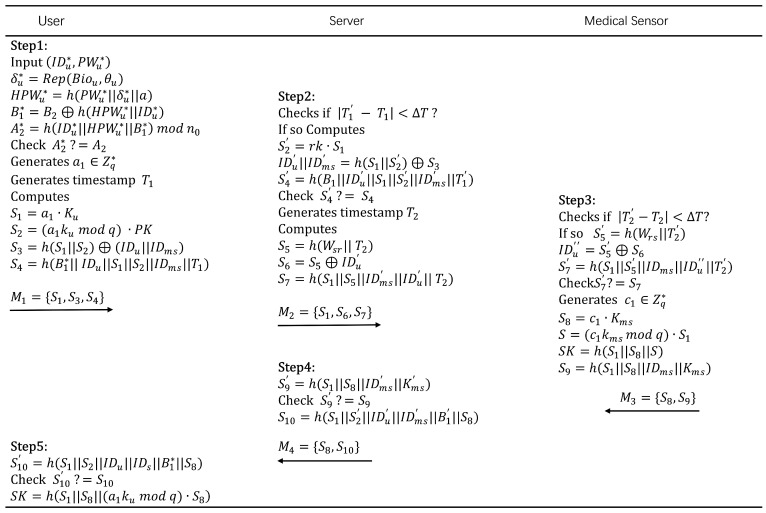
Authentication phase.

**Figure 5 sensors-25-06567-f005:**
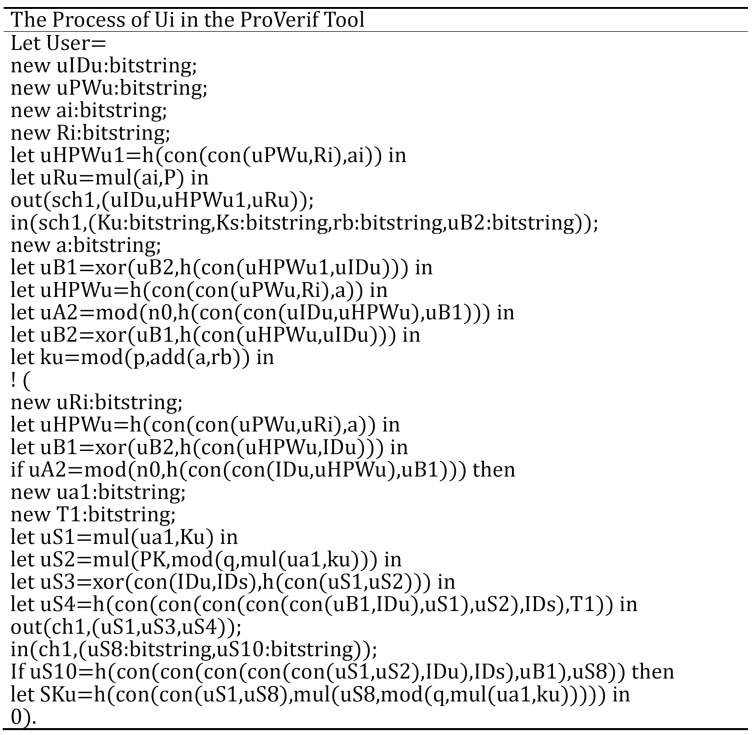
The process of $U_{i}$ in the ProVerif tool.

**Figure 6 sensors-25-06567-f006:**
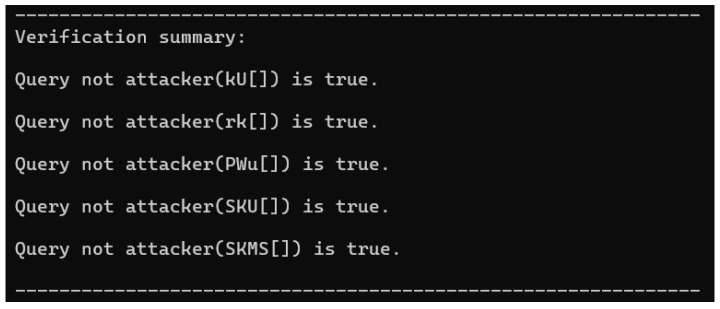
Simulation results.

**Figure 7 sensors-25-06567-f007:**
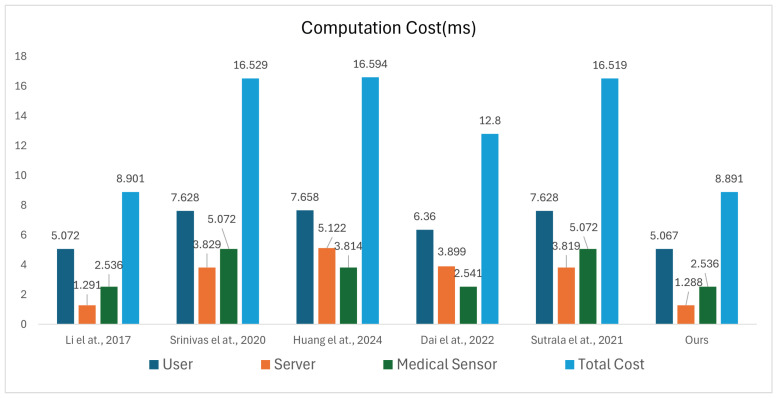
Simulation results [46,47,48,49,50].

**Figure 8 sensors-25-06567-f008:**
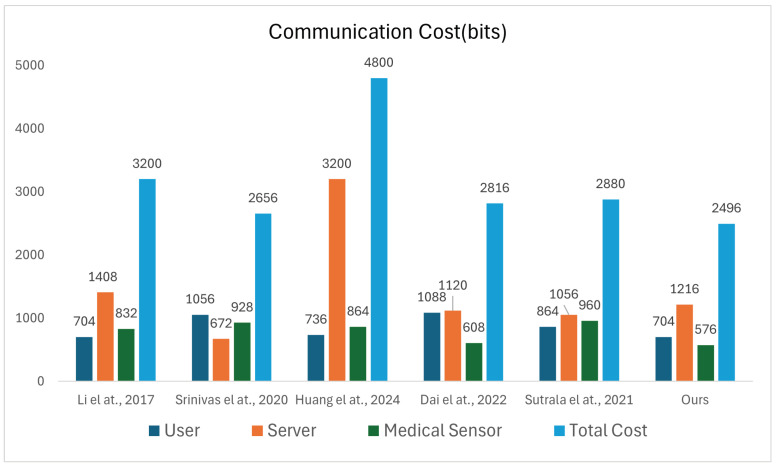
Simulation results [46,47,48,49,50].

**Figure 9 sensors-25-06567-f009:**
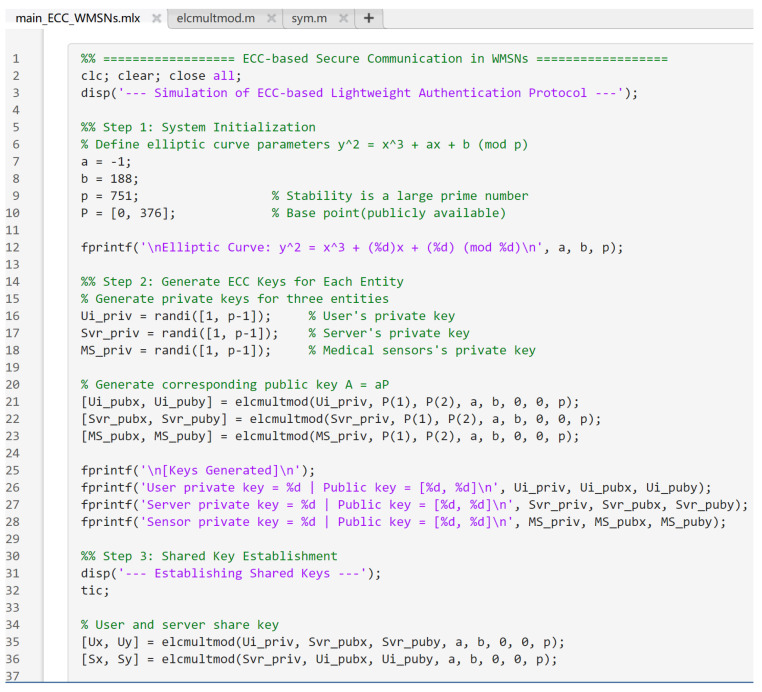
Partial code of simulation experiment.

**Figure 10 sensors-25-06567-f010:**
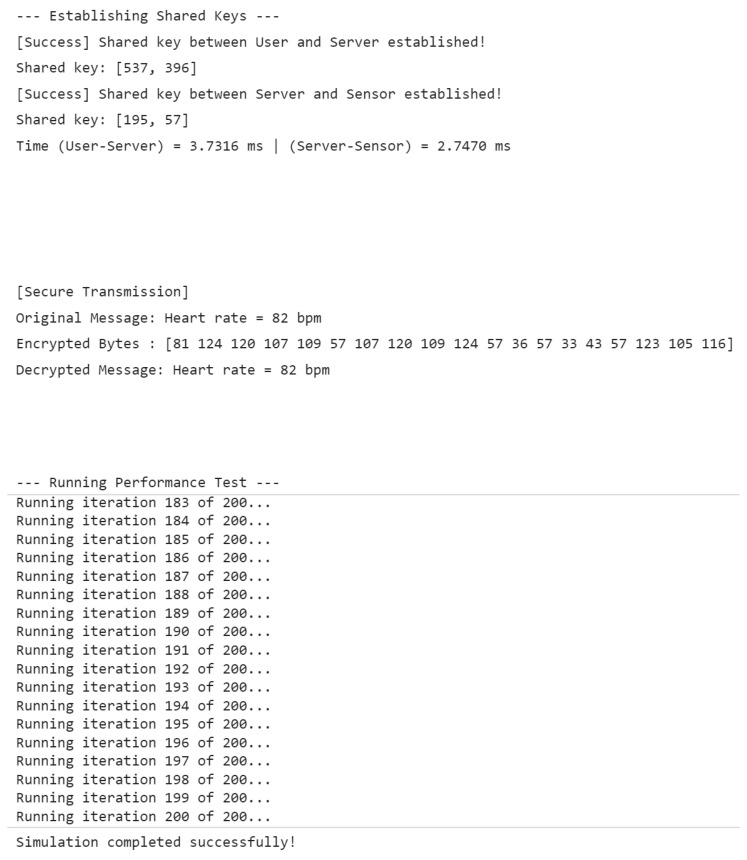
Simulation experiment results.

**Figure 11 sensors-25-06567-f011:**
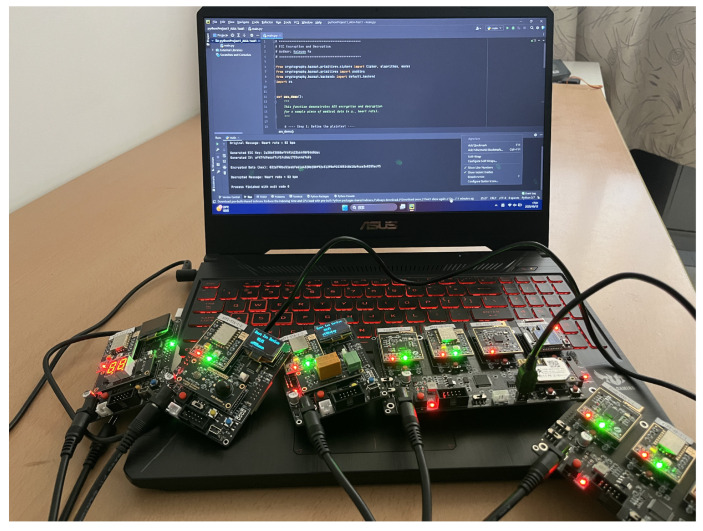
Experimental platform. Composed of five sensor nodes and a laptop.

**Table 1 sensors-25-06567-t001:** Notations and descriptions.

Notation	Description	Notation	Description
$U_{i}$	*i*th user	$MS_{j}$	*j*th medical sensor
$Svr_{k}$	*k*th server	A	an attacker
$SD_{u}$	user’s smart device	$Bio_{u}$	user’s biometric
$ID_{u}$	identity of $U_{i}$	$PW_{u}$	password of $U_{i}$
$ID_{ms}$	identity of $MS_{j}$	*T*	timestamp
$k_{u}$	private key of $U_{i}$	$K_{u}$	public key of $U_{i}$
$k_{ms}$	private key of $MS_{j}$	$K_{ms}$	public key of $MS_{j}$
rk	$Svr^{\prime }s$ private key	PK	$Svr^{\prime }s$ public key
SK	session key	$G_{en}/R_{ep}$	fuzzy extractor
→	public channel	⇢	security channel

**Table 2 sensors-25-06567-t002:** Evaluation criteria (C1–C12) and their definitions in WMSNs.

No.	Security Requirements	Definition in WMSNs
C1	No Password Verifier Table	$Srv_{k}$ doesn’t need to store the $U_{i}$’s password or the derived values of the $U_{i}$’s password.
C2	Password Friendly	$U_{i}$ is allowed to select their password and change it directly on $SD_{u}$.
C3	Session Key Agreement	Following the authentication process, a shared session key is generated between $U_{i}$ and $MS_{j}$ to enable secure communication.
C4	Mutual Authentication	$U_{i}$ and $Svr_{k}$, as well as $Svr_{k}$ and $MS_{j}$, can mutually authenticate each other’s authenticity.
C5	Sound Repairability	The scheme enables $U_{i}$ to revoke their $SD_{u}$ without altering their identities. Moreover, it allows for the dynamic integration of sensor nodes.
C6	User Anonymity	The scheme protects the $U_{i}$’s true identity, preventing the tracking of $U_{i}$ activities.
C7	Resistance to Known Attacks	The scheme is capable of defending against various known attacks, including user impersonation attacks, de-synchronization attacks, replay attacks, offline dictionary guessing attacks, and others.
C8	Resistance to Smart Device Loss Attacks	Even if A captures the smart device/card and extracts the parameters, they cannot recover the password nor use a password guessing attack to impersonate the user.
C9	Forward Secrecy	Leaking long-term keys will not impact the security of previous sessions.
C10	Resistance to Node Capture Attacks	A cannot compromise the protocol by capturing the medical sensor.
C11	Resistance Insider Attack	The legitimate user’s password information and session key SK cannot be directly accessed by the server, nor can it be obtained through simple computations.
C12	Resistance ESL Attack	In the scheme, even if the random numbers are leaked, the security of the protocol will not be compromised.

**Table 3 sensors-25-06567-t003:** Performance comparison.

Scheme	C1	C2	C3	C4	C5	C6	C7	C8	C9	C10	C11	C12
[46]	✓	✓	✓	✓	✓	✓	✓	×	✓	×	×	×
[47]	✓	✓	✓	×	×	×	×	×	✓	×	✓	×
[48]	✓	✓	✓	✓	✓	✓	×	✓	✓	✓	✓	×
[49]	✓	×	✓	✓	✓	✓	×	×	×	✓	✓	×
[50]	✓	✓	✓	✓	×	×	×	×	×	✓	✓	×
Ours	✓	✓	✓	✓	✓	✓	✓	✓	✓	✓	✓	✓

C1: no password verifier table; C2: password friendly; C3: session key agreement; C4: mutual authentication; C5: sound repairability; C6: user anonymity; C7: resists known attacks (user impersonation attacks, de-synchronization attacks, replay attacks, offline dictionary guessing attacks); C8: resists smart card loss attack; C9: forward secrecy; C10: resist node capture attack; C11: resists insider attack; C12: resists ESL attack. ✓: supporting a functional feature or ensuring security; ×: lacking a functional feature or not ensuring security.

**Table 4 sensors-25-06567-t004:** Computation costs.

Scheme	User (ms)	Server (ms)	Medical Sensor (ms)	Total Cost (ms)	Total (↓%)
[46]	$8T_{h}+3T_{m}+T_{B}\approx 5.072$	$7T_{h}+T_{m}\approx 1.291$	$4T_{h}+2T_{m}\approx 2.536$	$19T_{h}+6T_{m}+T_{B}\approx 8.901$	0.11%
[47]	$16T_{h}+5T_{m}+T_{B}\approx 7.628$	$11T_{h}+3T_{m}\approx 3.829$	$48T_{h}+4T_{m}\approx 5.072$	$35T_{h}+12T_{m}+T_{B}\approx 16.529$	46.2%
[48]	$22T_{h}+5T_{m}+T_{B}\approx 7.658$	$18T_{h}+4T_{m}\approx 5.122$	$8T_{h}+3T_{m}\approx 3.814$	$48T_{h}+12T_{m}+T_{B}\approx 16.594$	46.42%
[49]	$14T_{h}+4T_{m}+T_{B}\approx 6.36$	$25T_{h}+3T_{m}\approx 3.899$	$5T_{h}+2T_{m}\approx 2.541$	$44T_{h}+9T_{m}+T_{B}\approx 12.8$	30.54%
[50]	$16T_{h}+5T_{m}+T_{B}\approx 7.628$	$9T_{h}+3T_{m}\approx 3.819$	$8T_{h}+4T_{m}\approx 5.072$	$33T_{h}+12T_{m}+T_{B}\approx 16.519$	46.18%
Ours	$7T_{h}+3T_{m}+T_{B}\approx 5.067$	$6T_{h}+T_{m}\approx 1.288$	$4T_{h}+2T_{m}\approx 2.536$	$17T_{h}+6T_{m}+T_{B}\approx 8.891$	-

↓: reduction in the overhead of our scheme over existing schemes.

**Table 5 sensors-25-06567-t005:** Comparison of server computation costs.

Scheme	Server (ms)	Total (↓%)
[46]	$7T_{h}+T_{m}\approx 1.291$	0.23%
[47]	$11T_{h}+3T_{m}\approx 3.829$	66.36%
[48]	$18T_{h}+4T_{m}\approx 5.122$	74.89%
[49]	$25T_{h}+3T_{m}\approx 3.899$	66.98%
[50]	$9T_{h}+3T_{m}\approx 3.819$	66.27%
Ours	$6T_{h}+T_{m}\approx 1.288$	-

↓: reduction in the overhead of our scheme over existing schemes.

**Table 6 sensors-25-06567-t006:** Communication Costs.

Scheme	N.	User (Bit)	Server (Bit)	Medical Sensor (Bit)	Total Cost (Bit)	Total (↓%)
[46]	4	M+2ID+H	2M+2H+R	M+2H	3200	22%
[47]	3	2M+2ID+T+H	M+ID+T+H	M+ID+T+2H	2656	6.1%
[48]	4	M+2ID+H+T	3M+2ID+6H+2R+2T	M+2H+T	4800	48.85%
[49]	4	2M+2ID+2T+H	M+2R+T+H	M+T+H	2816	11.36%
[50]	4	M+4ID+R+T	M+2ID+H+R+3T	M+ID+H+R+2T	2880	13.33%
Ours	4	M+2ID+H	2M+ID+2H	M+H	2496	-

**N.**: denotes the number of communications. ↓: reduction in the overhead of our scheme over existing schemes.

**Table 7 sensors-25-06567-t007:** Description of experimental parameters.

Parameter	Description
Operating system	Windows 10/11
Cryptographic library	Micro-ECC
communication module	ESP-12F
Elliptic curves	Secp160r1, Secp192r1, Secp256k1
Hash function	SHA-256
Programming language	C (C Free 5)/Python 3.7

## Data Availability

Data are contained within the article. The original contributions presented in this study are included in the article. Further inquiries can be directed to the corresponding author.

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
