# Peer review of "A Secure and Lightweight ECC-Based Authentication Protocol for Wireless Medical Sensors Networks"

_sensors, 2025, doi:10.3390/s25216567_

Round 1

Reviewer 1 Report

Comments and Suggestions for Authors

please see the attached pdf file

Author Response

Please refer to the PDF file for the modified content.

Reviewer 2 Report

Comments and Suggestions for Authors

1. Currently, the paper has the following structure: Introduction, Related Work, Enhanced Attacker Model and Evaluation Criteria, Review of Wang et al’s Protocol, Proposed Protocol, Security Analysis, Performance Comparison, Conclusion. The paper should be reorganized to provide the following structure: Introduction, Materials and Methods, Results, Discussion, Conclusions. The structure is required by the journal template

https://www.mdpi.com/files/word-templates/sensors-template.dot

2. Describe where the obtained results can be applied at the end of Abstract and Conclusions. Addition of one sentence in Abstract and one sentence in Conclusions is quite enough.

3. Line 374. Remove a square sign in the end of sentence.

4. Line 415. Add a blank between ’Theorem’ and ’1’. It should be ’Theorem 1’ instead of ’Theorem1’.

5. Line 568. Add a reference number after ’Wang et al.’ in the sentence ’Security analysis of the protocol proposed by Wang et al. reveals that it fails to resist ESL attacks and gateway impersonation attacks.’ to avoid doubt or misunderstanding.

6. The future research plans and main numerical results should be added in Conclusions.

7. The references should be prepared exactly in accordance with the journal requirements. Journal names should be abbreviated. Missing DOIs or links should be added. References in recent manuscripts published in the journal could also help you with it.

8. Table header is missing in Table 2. It should be added.

9. Increase font size in plot Sensors in Figure 1.

10. Resolution should be increased in Figures 2, 3, 4, 5, 6, and 7.

Author Response

(The authors gave the same response as above.)

Reviewer 3 Report

Comments and Suggestions for Authors

This paper addresses the problem of insider privilege attacks and ephemeral secret leakage (ESL) attacks, which are often overlooked in Wireless Medical Sensor Networks (WMSNs), and proposes an ECC-based three-factor authentication protocol. By extending the attacker model and conducting formal verification under the Real-Or-Random (ROR) model and the ProVerif tool, the study theoretically enhances the security of the scheme and demonstrates certain novelty and academic value.
1. The fuzzy extractor for biometric features may be affected by noise and acquisition accuracy in real environments, which introduces a degree of instability.
2. In WMSNs, the privacy of images and physiological data is also a critical security challenge. The authors may refer to "A GAN-based anti-forensics method by modifying the quantization table in JPEG header file" and "Image privacy protection scheme based on high-quality reconstruction DCT compression and nonlinear dynamics." These studies enhance the security and imperceptibility of medical data from the perspective of image compression and anti-forensics, which are complementary to the ECC-based three-factor authentication protocol proposed in this paper, and may help extend the protocol’s capability in protecting medical image data transmission in future research.
3. The protocol mainly targets a single-server scenario and lacks support for large-scale multi-gateway medical IoT environments.
4. Since the protocol is based on ECC, it may face security threats in the future under quantum computing environments.
5. The current study imposes high security requirements on random numbers and session keys. The authors may consider incorporating the results of "A Three-Dimensional Memristor-Based Hyperchaotic Map for Pseudorandom Number Generation and Multi-Image Encryption" and "Encrypt a Story: A Video Segment Encryption Method Based on the Discrete Sinusoidal Memristive Rulkov Neuron." These works employ chaotic systems and memristive neurons to achieve high-quality key generation and video segment encryption, which could enhance the applicability of the protocol in multimodal medical data scenarios such as imaging and video monitoring.
6. The fluency of language expression should be improved, and redundant or repetitive descriptions should be avoided.
7. The conclusion section could benefit from the addition of future research directions, such as the integration with blockchain or trusted hardware.

Author Response

(The authors gave the same response as above.)

Reviewer 4 Report

Comments and Suggestions for Authors

This paper proposes a secure and lightweight ECC-based authentication protocol tailored for Wireless Medical Sensor Networks (WMSNs). As WMSNs play an increasingly important role in smart healthcare, ensuring the security of sensor data is a critical research topic. The proposed protocol enhances resistance against internal privileged attacks and ephemeral secret leakage (ESL) attacks through multi-factor authentication, combining biometric factors, random numbers, and user passwords. By analyzing the security of existing protocols, the paper demonstrates that the proposed protocol offers enhanced security while reducing computational and communication overhead. Compared with existing protocols, the proposed scheme achieves better performance in terms of both security and efficiency.

1. While the paper extends the attacker model to consider internal privileged attacks and ESL attacks, further analysis in more complex attack scenarios (such as joint attacks or multiple attackers) would be beneficial. Simulating combinations of attacks could further validate the robustness of the protocol.

2. Although the paper compares the proposed protocol with several existing schemes, the datasets used in the experiments are limited. It is recommended to include more real-world scenarios to validate the protocol’s effectiveness. For instance, testing in different network scales, various medical sensor types, and challenging network environments would provide a comprehensive performance evaluation.

3. The paper mentions the formal verification of the protocol using the ProVerif tool but does not provide in-depth details on the verification process and results. It would be helpful to describe how ProVerif was used for verification, especially in different security assumptions, to help readers understand the tool's role in the process.

4. The experimental results show that the proposed protocol reduces computational and communication overhead compared to other protocols. However, further discussion on optimizing computational overhead, such as algorithm improvements or hardware acceleration, would be valuable, especially in large-scale network applications.

5. The paper outlines future research directions, but this could be expanded further. For example, discussing the potential integration of machine learning or AI techniques to dynamically adapt to different attack strategies or improve security in WMSNs could open new avenues for further study.

The motivation for writing this paper needs to be further strengthened to improve the content quality of the manuscript. Some existing work is helpful for the manuscript, such as a survey when moving target defense meets game theory.

Author Response

(The authors gave the same response as above.)

Reviewer 5 Report

Comments and Suggestions for Authors
  1. Authors should be able to grasp the whole scape of one paper from its abstract. However, authors spend more than sixty percent of abstract to address the background or motivation of this study. Authors might outline the more essential parts of this study such as the methodology, contribution, or a brief conclusion, from which to gain attention for readers to read the contents. Thus, the entire abstract should be rewritten to match basic requirements.
  2. In this paper, a secure and lightweight ECC-based authentication protocol for wireless medical sensors networks (WMSNs) is proposed. At first, the topic of authentication protocol for WMSNs might not provide much novelty. Secondly, from the structure of this manuscript, sections 3 and 4 address the enhanced attacker model and evaluation criteria and review of Wang et al’s protocol, respectively, and section 5 provides a detailed explanation of the phases of the proposed protocol. Thus, these parts cannot provide much contribution. The essential part of this study, which might provide contribution to this study, should be section 6 entitled --- a security analysis and experimental evaluation. However, as addressed in section 6.2, authors can provide only descriptive security analysis. Readers would expect sound quantitative results or more theoretical analysis to support the proposed scheme, instead of that of Automatic Formal Verification by ProVerif in section 6.3. In other words, this study might not provide sound contributions.
  3. Performance comparison of the proposed protocol with that of related protocols [39–43], based on emphasizing security features, computational efficiency, and communication overhead. Providing more detailed data as well as detailed explanations of performance comparison is preferable.
  4. The entire contents might not be able to match the title of paper, which is a secure and lightweight ECC-based authentication protocol for wireless medical sensors networks. Readers might expect more ECC-related work to appear in the contents from this title.

Author Response

(The authors gave the same response as above.)

Reviewer 6 Report

Comments and Suggestions for Authors
  • Figures 2–4 (showing different protocol phases) could be made clearer with better labeling and step numbering.
  •  Although the literature review is comprehensive, it could be made more analytical. The authors could explicitly highlight how each existing protocol fails to address insider or ESL attacks rather than just listing them sequentially.
  • While computational costs are presented, the experimental setup is somewhat brief. Providing more information on parameter choices (e.g., ECC curve type, key length, cryptographic library used) would improve reproducibility.
  • The ROR-based proof is mathematically sound, but some transitions between games (GM0–GM5) are not intuitively explained.
  • There are minor typographical inconsistencies (spacing before punctuation, capitalization of section titles, etc.).
  • Section 5.4 (Password Change Phase): A brief explanation of why local password change is secure would strengthen this part.

Author Response

(The authors gave the same response as above.)

Round 2

Reviewer 1 Report

Comments and Suggestions for Authors

The authors have precisely addressed my recommendations. The paper has been significantly improved and can be accepted. 

Reviewer 3 Report

Comments and Suggestions for Authors

I really appreciate the authors' revisions, which effectively address the issues that were raised.

Reviewer 5 Report

Comments and Suggestions for Authors

Authors make a great effort to improve the quality of the manuscript and provide the detailed explanations how the revised manuscript is modified to answer my comments, and there is no further comments to the revised manuscript.

Reviewer 6 Report

Comments and Suggestions for Authors

The authors have satisfactorily revised the manuscript and addressed all previously raised concerns and comments. No further revisions are required.